# $SO(n)$ Affleck-Kennedy-Lieb-Tasaki states as conformal boundary states of integrable $SU(n)$ spin chains

**Yueshui Zhang[1], Ying-Hai Wu[2], Meng Cheng[3*], Hong-Hao Tu[1†]**

**1** Faculty of Physics and Arnold Sommerfeld Center for Theoretical Physics,
Ludwig-Maximilians-Universität München, 80333 Munich, Germany
**2** School of Physics and Wuhan National High Magnetic Field Center, Huazhong University of
Science and Technology, Wuhan 430074, China
**3** Department of Physics, Yale University, New Haven, CT 06511-8499, USA

⋆ m.cheng@yale.edu , † h.tu@lmu.de

## Abstract

We construct a class of conformal boundary states in the $SU(n)_1$ Wess-Zumino-Witten
(WZW) conformal field theory (CFT) using the symmetry embedding $Spin(n)_2 \subset SU(n)_1$.
These boundary states are beyond the standard Cardy construction and possess $SO(n)$
symmetry. The $SU(n)$ Uimin-Lai-Sutherland (ULS) spin chains, which realize the $SU(n)_1$
WZW model on the lattice, allow us to identify these boundary states as the ground states
of the $SO(n)$ Affleck-Kennedy-Lieb-Tasaki spin chains. Using the integrability of the $SU(n)$
ULS model, we analytically compute the corresponding Affleck-Ludwig boundary entropy
using exact overlap formulas. Our results unveil intriguing connections between exotic
boundary states in CFT and integrable lattice models, thus providing deep insights into
the interplay of symmetry, integrability, and boundary critical phenomena.

# 1  Introduction

Conformal boundary conditions play a central role in two-dimensional conformal field theories (CFTs), with wide applications in condensed matter physics and string theory. In rational CFTs, a distinguished class of boundary states, known as Cardy states, can be systematically constructed using modular data and the fusion algebra [1–6]. While Cardy's construction accounts for all boundary conditions in many minimal models, it is now understood that additional, non-Cardy boundary states can arise more generally [7, 8], for example, through topological defect lines (TDLs), orbifold constructions, or symmetry embeddings. A complete classification of conformal boundary states remains an open issue, even in well-understood rational CFTs.

One general mechanism for generating such non-Cardy boundary states is via conformal embeddings, where a CFT with symmetry group $G$ is embedded into another CFT with a larger symmetry group $G'$. This allows one to construct conformal boundary states that preserve only $G$ for the CFT with symmetry $G'$ [8], which cannot be obtained by acting with the usual Verlinde lines. The resulting boundary states are beyond the Cardy scheme and can exhibit symmetry-breaking patterns or projective representations not accessible from the original fusion algebra. In particular, TDLs associated with the embedded theory can be used to generate new boundary conditions when applied to standard Cardy states of the parent theory. These constructions provide a systematic route to realizing conformal boundary conditions with reduced symmetry, which naturally arise in settings involving symmetry breaking, gauging, or topological defects.

Although this approach yields a rich class of conformal boundary states, many fundamental questions remain unsolved. In particular, it is often unclear how to realize these boundary conditions in microscopic models. Establishing explicit connections between these boundary states and integrable systems would not only provide quantitative access to their universal properties, but also enable a non-perturbative analysis of their renormalization group behavior.

In this work, we propose a class of non-Cardy conformal boundary states in the $SU(n)_1$ Wess-Zumino-Witten (WZW) conformal field theory that arises from the conformal embedding $Spin(n)_2 \subset SU(n)_1$. These boundary states are symmetric only under the $SO(n)$ subgroup of $SU(n)$ and can be constructed by applying non-invertible TDLs associated with the embedded $Spin(n)_2$ theory. We further identify their lattice realizations in the integrable $SU(n)$ Uimin-Lai-Sutherland (ULS) spin chain [9–11], whose low-energy limit is described by the $SU(n)_1$ WZW model [12]. We find that $SO(n)$-symmetric matrix product states (MPSs) [13, 14], specifically the ground states of generalized Affleck-Kennedy-Lieb-Tasaki (AKLT) spin chains [15], realize the aforementioned non-Cardy boundary states on the lattice. By employing Bethe ansatz techniques, we extract the Affleck-Ludwig boundary entropy [5] exactly from the overlap between the $SO(n)$ MPS and the ground state of the $SU(n)$ ULS chain, which perfectly matches the CFT prediction. Our work builds upon Affleck's seminal contributions to boundary CFT and AKLT wave functions, which established a framework for understanding gapless and gapped phases of matter in one dimension. Our results provide a concrete example of how non-Cardy boundary states can be realized and analyzed in lattice models using integrability methods.

# 2  Conformal embedding $Spin(n)_2 \subset SU(n)_1$

We construct non-Cardy boundary states in the $SU(n)_1$ WZW model by exploiting the conformal embedding $Spin(n)_2 \subset SU(n)_1$, which is possible because two theories have the same central charge $c = n - 1$.

## 2.1  Odd $n$

To illustrate the construction, we begin with the case of odd $n = 2r + 1$ ($r \geq 1$). The Spin$(2r+1)_2$ theory has $r + 4$ chiral primary fields, denoted as $I, Z, X, X', Y_m$ ($m = 1, \ldots, r$). Their conformal weights are

$$h_I = 0, \quad h_Z = 1, \quad h_X = \frac{r}{8}, \quad h_{X'} = \frac{r+4}{8}, \quad h_{Y_m} = \frac{m(n-m)}{2n}. \tag{1}$$

On the other hand, the SU$(n)_1$ theory has $n$ chiral primary fields labeled by $[a]$, $a = 0, \ldots, n-1$, with conformal weights $h_{[a]} = a(n-a)/2n$. The embedding manifests itself through the partition function on the torus: the diagonal modular invariant of SU$(2r+1)_1$ coincides with the following off-diagonal modular invariant of Spin$(2r+1)_2$:

$$\mathcal{Z} = |\chi_I + \chi_Z|^2 + 2 \sum_{m=1}^{r} |\chi_{Y_m}|^2, \tag{2}$$

where $\chi$'s are chiral characters of the CFT. The characters of the two CFTs are identified via

$$\chi_I + \chi_Z = \chi_{[0]}, \quad \chi_{Y_m} = \chi_{[m]} = \chi_{[2r+1-m]}, \quad m = 1, \ldots, r. \tag{3}$$

Notably, the fields $X$ and $X'$ do not appear in the partition function [Eq. (2)]. Nevertheless, they can be used to define a simple TDL, $\mathcal{D}_\sigma = \frac{1}{2}(\mathcal{D}_X + \mathcal{D}_{X'})$, whose fusion rule reads (see Appendix A)

$$\mathcal{D}_\sigma \times \mathcal{D}_\sigma = \sum_{a=0}^{2r} \mathcal{D}_{[a]}, \tag{4}$$

where $\mathcal{D}_X$, $\mathcal{D}_{X'}$ and $\mathcal{D}_{[a]}$ denote the Verlinde lines in Spin$(2r+1)_2$ and SU$(2r+1)_1$ theories, respectively. Acting on the SU$(2r+1)_1$ identity Cardy state $|[0]\rangle$, this TDL yields a new conformal boundary state for SU$(2r+1)_1$:

$$|\mathcal{D}_\sigma\rangle = \mathcal{D}_\sigma |[0]\rangle = \frac{1}{\sqrt{2}} \left( |X\rangle + |X'\rangle \right), \tag{5}$$

where $|X\rangle$ and $|X'\rangle$ are Cardy states in the Spin$(2r+1)_2$ theory. The fusion rule in Eq. (4) implies that the TDL $\mathcal{D}_\sigma$ is self-conjugate, and thus the non-Cardy state $|\mathcal{D}_\sigma\rangle$ is invariant under charge conjugation.

The corresponding partition function on a cylinder, with boundary states $|\mathcal{D}_\sigma\rangle$ placed at two ends, is defined as

$$\mathcal{Z}_{\mathcal{D}_\sigma, \mathcal{D}_\sigma} = \langle \mathcal{D}_\sigma | e^{-\beta H} | \mathcal{D}_\sigma \rangle, \tag{6}$$

where $H = \frac{2\pi}{L}(L_0 + \bar{L}_0 - \frac{c}{12})$ is the SU$(2r+1)_1$ CFT Hamiltonian on a circle of length $L$, and $\beta$ is the inverse temperature. In the low-temperature limit ($\beta \to \infty$), the dominant contribution comes from the vacuum state $|0\rangle$, yielding

$$\mathcal{Z}_{\mathcal{D}_\sigma, \mathcal{D}_\sigma} \simeq |\langle 0 | \mathcal{D}_\sigma \rangle|^2 \, e^{\frac{\pi c \beta}{6L}}. \tag{7}$$

Alternatively, using the fusion rule [Eq. (4)], the same partition function can be written as

$$\mathcal{Z}_{\mathcal{D}_\sigma, \mathcal{D}_\sigma} = \sum_{a=0}^{2r} \chi_{[a]}(q) \tag{8}$$

with $q = e^{-\pi L/\beta}$. Applying a modular $S$ transformation to the SU$(n)_1$ characters in Eq. (8), $\chi_{[a]}(q) = \sum_{b=0}^{n-1} S_{ab} \chi_{[b]}(q')$ with $S_{ab} = \frac{1}{\sqrt{n}} e^{2\pi i ab/n}$ and $q' = e^{-4\pi\beta/L}$, one extracts the Affleck-Ludwig $g$ factor:

$$g_{\mathcal{D}_\sigma} \equiv \langle 0 | \mathcal{D}_\sigma \rangle = (2r+1)^{1/4} = n^{1/4}, \tag{9}$$

which quantifies the universal boundary entropy [5] associated with the non-Cardy state $|\mathcal{D}_\sigma\rangle$.

## 2.2 Even $n$

We now turn to the case of even $n = 2r + 2$ ($r \geq 1$). The Spin$(2r+2)_2$ theory has $r+8$ primary fields, denoted as $I$, $Z$, $X_\pm$, $X'_\pm$, $W_\pm$, and $Y_m$ ($m = 1, \ldots, r$). Their conformal weights are

$$h_I = 0, \quad h_Z = 1, \quad h_{X_\pm} = \frac{2r+1}{16}, \quad h_{X'_\pm} = \frac{2r+9}{16}, \quad h_{W_\pm} = \frac{r+1}{4}, \quad h_{Y_m} = \frac{m(n-m)}{2n}. \quad (10)$$

The embedding relates the diagonal partition function of SU$(2r+2)_1$ to the following off-diagonal modular invariant of Spin$(2r+2)_2$:

$$\mathcal{Z} = |\chi_I + \chi_Z|^2 + 2\sum_{m=1}^{r} |\chi_{Y_m}|^2 + |\chi_{W_+} + \chi_{W_-}|^2 \quad (11)$$

with the identification of characters

$$\chi_I + \chi_Z = \chi_{[0]}, \quad \chi_{W_+} + \chi_{W_-} = \chi_{[r+1]}, \quad \chi_{Y_m} = \chi_{[m]} = \chi_{[2r+2-m]}, \quad m = 1, \ldots, r. \quad (12)$$

The Verlinde lines corresponding to the fields $X_\pm$ and $X'_\pm$ can be used to define two simple TDLs in SU$(2r+2)_1$, $\mathcal{D}_{\sigma_\pm} = \frac{1}{2}(\mathcal{D}_{X_\pm} + \mathcal{D}_{X'_\pm})$, with fusion rules (see Appendix A):

$$\mathcal{D}_{\sigma_\pm} \times \mathcal{D}_{\sigma_\pm} = \sum_{\substack{a=0\,(a\,\text{even})}}^{2r+1} \mathcal{D}_{[a]}, \quad \text{odd } r,$$

$$\mathcal{D}_{\sigma_+} \times \mathcal{D}_{\sigma_-} = \sum_{\substack{a=0\,(a\,\text{even})}}^{2r+1} \mathcal{D}_{[a]}, \quad \text{even } r. \quad (13)$$

Acting each of these TDLs on the identity Cardy state gives a non-Cardy boundary state:

$$|\mathcal{D}_{\sigma_\pm}\rangle = \mathcal{D}_{\sigma_\pm}|[0]\rangle = \frac{1}{\sqrt{2}}\left(|X_\pm\rangle + |X'_\pm\rangle\right), \quad (14)$$

where $|X_\pm\rangle$ and $|X'_\pm\rangle$ are Cardy states in the Spin$(2r+2)_2$ theory. The fusion rules in Eq. (13) imply that $|\mathcal{D}_{\sigma_\pm}\rangle$ are both invariant under the charge conjugation for odd $r$, while they are conjugate to each other for even $r$. Accordingly, the cylinder partition function with boundary states $|\mathcal{D}_{\sigma_\pm}\rangle$ at two ends reads

$$\mathcal{Z}_{\mathcal{D}_{\sigma_\pm}, \mathcal{D}_{\sigma_\pm}} = \langle \mathcal{D}_{\sigma_\pm}|e^{-\beta H}|\mathcal{D}_{\sigma_\pm}\rangle = \sum_{\substack{a=0\,(a\,\text{even})}}^{2r+1} \chi_{[a]}(q). \quad (15)$$

As in the odd $n$ case, a modular $S$ transformation gives the Affleck-Ludwig $g$ factor for $|\mathcal{D}_{\sigma_\pm}\rangle$:

$$g_{\mathcal{D}_{\sigma_\pm}} \equiv \langle 0|\mathcal{D}_{\sigma_\pm}\rangle = \frac{(2r+2)^{1/4}}{\sqrt{2}} = \frac{n^{1/4}}{\sqrt{2}}. \quad (16)$$

## 3 Integrable lattice realization

To realize the non-Cardy boundary states in microscopic models, we previously proposed a mechanism based on impurity screening [8]. Here, we take a complementary approach: after space-time rotation, conformal boundary states appear as physical states with short-range correlation [16]. Such states often emerge as ground states of gapped phases near a critical

point. This scenario is illustrated in Fig. 1, where a CFT is adjacent to gapped phases, described by certain massive field theories, at both ends. The correlation length of the massive theory is much shorter than the system size. In the infrared (IR) limit, the composite system is effectively described by a boundary CFT (BCFT), with the two interfaces flowing to conformal boundary conditions at the respective ends. The space-time rotation provides an imaginary-time evolution perspective, where the physical degrees of freedom of the gapped phases are projected to their ground states, thereby realizing regularized conformal boundary states.

A well-known example is the transverse-field Ising (TFI) chain $H = -\sum_j \sigma_j^z \sigma_{j+1}^z - h \sum_j \sigma_j^x$, which realizes the Ising CFT at $h = 1$. Its three Cardy boundary states, corresponding to the primary fields $I, \sigma, \varepsilon$, can be taken as $|I\rangle = |\uparrow\uparrow \cdots \uparrow\rangle$, $|\varepsilon\rangle = |\downarrow\downarrow \cdots \downarrow\rangle$, and $|\sigma\rangle = |+ + \cdots +\rangle$, where $\sigma^x|+\rangle = |+\rangle$. These states are exact ground states of the TFI chain at $h = 0$ and $h = \infty$, illustrating how gapped phases away from criticality encode conformal boundary conditions.

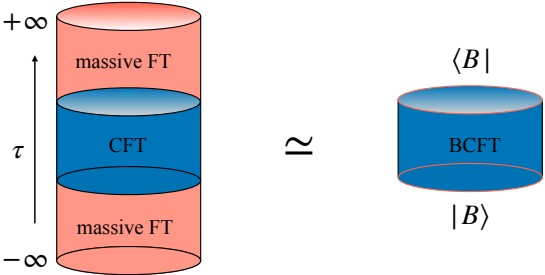

Figure 1: The imaginary-time evolution picture of a composite system consisting of a CFT coupled to massive field theories at both ends (left), and the effective BCFT description in the IR limit (right), where the ground state of the massive field theory flows to the conformal boundary state $|B\rangle$.

Motivated by this picture, we consider the following one-dimensional SO($n$)-symmetric bilinear-biquadratic spin chain [13, 14] as a natural setting for realizing non-Cardy boundary states discussed above:

$$H = \sum_{j=1}^{N} \left[ \cos\theta \sum_{a<b} T_j^{ab} T_{j+1}^{ab} + \sin\theta \left( \sum_{a<b} T_j^{ab} T_{j+1}^{ab} \right)^2 \right], \qquad (17)$$

where $T^{ab} = i(|a\rangle\langle b| - |b\rangle\langle a|)$ ($0 \le a < b \le n-1$) are the $n(n-1)/2$ generators of the SO($n$) Lie algebra in the vector representation, and $|a\rangle$ ($a = 0, 1, \ldots, n-1$) denote the $n$ local basis states at each site. We adopt periodic boundary condition, and $N$ is the total number of sites.

This model has an integrable point at $\theta = \arctan \frac{1}{n-2}$, known as the Uimin-Lai-Sutherland (ULS) point [9–11], where the SO($n$) symmetry is enhanced to PSU($n$). The low-energy effective theory at this point is the SU($n$)$_1$ WZW model [12]. The ground state of the ULS chain, denoted as $|\psi_0\rangle$, corresponds to the CFT vacuum and will serve as a reference state for evaluating overlaps with trial boundary states.

Away from the ULS point, the PSU($n$) symmetry is explicitly broken down to SO($n$), and the model enters a gapped phase when $\theta < \arctan \frac{1}{n-2}$. From the field-theory viewpoint, the gap generation can be understood as perturbing the SU($n$)$_1$ WZW model by marginally relevant terms that only preserve SO($n$) symmetry [17]. The gapped phase terminates at another integrable critical point $\theta = \arctan \frac{n-4}{(n-2)^2}$, known as the Reshetikhin point [18]. It has been argued that the critical theory at this point is the SO($n$)$_1$ WZW model [13], and the low-energy effective theory in the regime $\theta \in (\arctan \frac{n-4}{(n-2)^2}, \arctan \frac{1}{n-2})$ is described by the SO($n$)$_1$ WZW model with a relevant perturbation [19, 20]. Of particular interest is the point $\theta = \arctan \frac{1}{n}$ [13, 14], where the ground states have exact MPS representations, generalizing

the celebrated spin-1 AKLT chain [15]. These gapped states are natural candidates for the non-Cardy conformal boundary states mentioned earlier. For odd (even) $n$ cases, the ground state at the MPS point is unique (two-fold degenerate), written as

$$|\Psi\rangle = |\text{MPS}\rangle, \quad |\Psi_\pm\rangle = \frac{1}{\sqrt{2}}\left(|\text{MPS}\rangle \pm |\widetilde{\text{MPS}}\rangle\right) \tag{18}$$

with

$$|\text{MPS}\rangle = \frac{1}{\sqrt{\mathcal{N}}} \sum_{a_1,\dots,a_N=0}^{n-1} \text{Tr}(\Gamma^{a_1}\dots\Gamma^{a_N})|a_1\dots a_N\rangle,$$

$$|\widetilde{\text{MPS}}\rangle = \frac{1}{\sqrt{\mathcal{N}}} \sum_{a_1,\dots,a_N=0}^{n-1} \text{Tr}(\Gamma^n\Gamma^{a_1}\dots\Gamma^{a_N})|a_1\dots a_N\rangle, \tag{19}$$

respectively. Here, $\Gamma^a$ are Gamma matrices satisfying the Clifford algebra $\{\Gamma^a, \Gamma^b\} = 2\delta_{ab}$, and $\mathcal{N}$ is the normalization constant. For even $n$, $\Gamma^n$ is defined as

$$\Gamma^n = (-i)^{n/2}\Gamma^0\Gamma^1\cdots\Gamma^{n-1}. \tag{20}$$

Recently, it has been pointed out that the two-fold degeneracy in the even $n$ cases can be understood as a consequence of the spontaneous breaking of a $\mathbb{Z}_2$ symmetry [21,22]. It is known that there exists a charge conjugation matrix $C$ that maps the SO($n$) spinor representation to its charge conjugation representation via $C\Gamma^a C^{-1} = -(\Gamma^a)^*$ [23], where $(\Gamma^a)^*$ denotes complex conjugation. Consequently, the AKLT states in Eq. (18) are not only SO($n$)-symmetric, but also transform under charge conjugation in the same way as the non-Cardy states in Eqs. (5) and (14), namely, $|\Psi^*\rangle = |\Psi\rangle$ for odd $n$, and $|\Psi_\pm^*\rangle = |\Psi_\mp\rangle$ for even $n$.

Taking these considerations into account, we propose identifying the MPSs in Eq. (18) as lattice realizations of the non-Cardy conformal boundary states discussed above:

$$\begin{aligned}
|\Psi\rangle &\rightarrow |\mathcal{D}_\sigma\rangle, &&\text{odd } n, \\
|\Psi_\pm\rangle &\rightarrow |\mathcal{D}_{\sigma_\pm}\rangle, &&\text{even } n.
\end{aligned} \tag{21}$$

In particular, one can easily verify that the lattice momentum is 0 for $|\text{MPS}\rangle$, and $\pi$ for $|\widetilde{\text{MPS}}\rangle$. For even $n$, both are non-injective. They should be understood as Schrödinger cat-like states that spontaneously break the lattice translation symmetry. We thus form the symmetric and anti-symmetric combinations $|\Psi_\pm\rangle$, which are short-range correlated and can be used as physical boundary states.

Remarkably, recent developments in integrability techniques have demonstrated that $|\text{MPS}\rangle$ is an *integrable* boundary state of the ULS chain, with integrability ensured by the so-called $KT$-relation [24–27]. Furthermore, since the $K$-matrix given in Ref. [26] commutes with the additional Gamma matrix $\Gamma^n$ for even $n$, it follows that $|\widetilde{\text{MPS}}\rangle$ is also an integrable boundary state. In this sense, we conclude that the SO($n$) AKLT states in Eq. (18) are integrable boundary states of the SU($n$) ULS chain for both odd and even $n$. The integrability makes it possible to analyze the overlap between the SO($n$) AKLT states and ULS-chain eigenstates rigorously.

Let us denote the Bethe eigenstates of the SU($n$) ULS chain as $|\{u\}\rangle$, where $\{u\}$ consists of Bethe roots $u_k^{(a)}$ ($a = 1,\dots,n-1$ and $k = 1,\dots,M_a$ with $0 \le M_{n-1} \le \cdots \le M_1 \le N$). These Bethe roots satisfy the nested Bethe Ansatz equations (BAEs) [11]:

$$\prod_{l=1}^{M_{a-1}} \frac{u_k^{(a)} - u_l^{(a-1)} + \frac{i}{2}}{u_k^{(a)} - u_l^{(a-1)} - \frac{i}{2}} = \prod_{l\neq k}^{M_a} \frac{u_k^{(a)} - u_l^{(a)} + i}{u_k^{(a)} - u_l^{(a)} - i} \prod_{l=1}^{M_{a+1}} \frac{u_j^{(a)} - u_l^{(a+1)} - \frac{i}{2}}{u_j^{(a)} - u_l^{(a+1)} + \frac{i}{2}}. \tag{22}$$

Taking the logarithm of both sides and introducing the associated Bethe quantum numbers $\{I_k^{(a)}\}$, the BAEs can be rewritten as

$$\frac{2\pi}{N}I_k^{(a)} = \phi_N^{(a)}(u_k^{(a)}),\tag{23}$$

where

$$\phi_N^{(a)}(u) = \vartheta_1(u)\delta_{a,1} + \frac{1}{N}\sum_{l=1}^{M_{a-1}}\vartheta_1(u-u_l^{(a-1)}) - \frac{1}{N}\sum_{l=1}^{M_a}\vartheta_2(u-u_l^{(a)}) + \frac{1}{N}\sum_{l=1}^{M_{a+1}}\vartheta_1(u-u_l^{(a+1)}),\tag{24}$$

with $\vartheta_p(u) = 2\arctan(2u/p)$ for $p = 1, 2$. The set of auxiliary functions $\phi_N^{(a)}(u)$ are commonly referred to as counting functions in the literature [28].

For simplicity, we set the chain length to be multiples of $2n$, i.e., $\mathrm{mod}(N, 2n) = 0$, which ensures that the ULS-chain ground state $|\psi_0\rangle$ always has zero lattice momentum for both odd and even $n$ [29]. As $|\mathrm{MPS}\rangle$ and $|\widetilde{\mathrm{MPS}}\rangle$ have lattice momenta $0$ and $\pi$, respectively, $|\psi_0\rangle$ has a vanishing overlap with $|\widetilde{\mathrm{MPS}}\rangle$, and it is sufficient to consider the overlap $\langle\psi_0|\mathrm{MPS}\rangle$. For any Bethe eigenstate $|u\rangle$ with vanishing lattice momentum, Bethe roots appear in pairs, $\{u_k^{(a)}\} = \{u_{k_+}^{(a)}\} \cup \{-u_{k_+}^{(a)}\}$. Remarkably, for such Bethe eigenstates (including the ground state $|\psi_0\rangle$), an exact formula for the overlap with $|\mathrm{MPS}\rangle$ is available [30–32] (see also Appendix B):

$$\langle\{u\}|\mathrm{MPS}\rangle = \frac{2^{\left[\frac{n}{2}\right]}}{\sqrt{\mathcal{N}}}\sqrt{\prod_{a=1}^{n-1}\prod_{k_+=1}^{M_a/2}\frac{(u_{k_+}^{(a)})^2 + 1/4}{(u_{k_+}^{(a)})^2}}\sqrt{\frac{\det G_N^+}{\det G_N^-}},\tag{25}$$

where $[n/2]$ denotes the integer part of $n/2$, and $G_N^\pm$ are factorized Gaudin matrices with indices running over the positive Bethe roots, defined by

$$[G_N^\pm]_{ab;k_+l_+} = 2\pi N\rho_N^{(a)}(u_{k_+}^{(a)})\delta_{ab}\delta_{k_+l_+} + K_{ab}(u_{k_+}^{(a)} - u_{l_+}^{(b)}) \pm K_{ab}(u_{k_+}^{(a)} + u_{l_+}^{(b)}),\tag{26}$$

with

$$\rho_N^{(a)}(u) = \frac{1}{2\pi}\frac{\mathrm{d}}{\mathrm{d}u}\phi_N^{(a)}(u),$$
$$K_{ab}(u) = \delta_{ab}\frac{\mathrm{d}}{\mathrm{d}u}\vartheta_2(u) - (\delta_{a+1,b} + \delta_{a-1,b})\frac{\mathrm{d}}{\mathrm{d}u}\vartheta_1(u).\tag{27}$$

# 4 Extracting the Affleck-Ludwig entropy

In this section, we demonstrate how the overlap formula in Eq. (25) allows us to extract the Affleck-Ludwig boundary entropy, thus confirming that the $SO(n)$ AKLT states are indeed lattice realizations of the non-Cardy conformal boundary states [Eq. (21)] for the integrable $SU(n)$ ULS chains.

For this purpose, we compute the overlaps

$$\langle\psi_0|\Psi\rangle = \langle\psi_0|\mathrm{MPS}\rangle, \quad \text{odd } n,$$
$$\langle\psi_0|\Psi_\pm\rangle = \frac{1}{\sqrt{2}}\langle\psi_0|\mathrm{MPS}\rangle, \quad \text{even } n,\tag{28}$$

where $|\psi_0\rangle$ is the ground state of the $SU(n)$ ULS chain. The overlap $\langle\psi_0|\mathrm{MPS}\rangle$ is expected to have the following asymptotic behavior:

$$\ln|\langle\psi_0|\mathrm{MPS}\rangle| = -\alpha N + \ln g + \cdots,\tag{29}$$

where $\alpha$ is a non-universal constant (often referred to as the "surface free energy density") and $\ln g$ is a universal subleading term related to the Affleck-Ludwig entropy [5]. For both odd and even $n$ cases, we expect $g = n^{1/4}$ [$g = g_{\mathcal{D}\sigma}$ for odd $n$ and $g = \sqrt{2}g_{\mathcal{D}\sigma\pm}$ for even $n$; see Eqs. (9) and (16)]. The ellipsis "$\cdots$" in Eq. (29) denotes finite-size corrections that vanish in the thermodynamic limit.

For the chain length under consideration $(\mathrm{mod}(N, 2n) = 0)$, the Bethe quantum numbers of the ULS-chain ground state $|\psi_0\rangle$ are given by $I_k^{(a)} = k - (M_a - 1)/2$, with $M_a = (n-a)N/n$, which uniquely solve the Bethe roots via the BAEs [Eq. (23)]. By inserting the Bethe roots into the overlap formula [Eq. (25)], the logarithm of the overlap can be decomposed into three terms:

$$\ln|\langle\psi_0|\mathrm{MPS}\rangle| = \left(\left[\frac{n}{2}\right]\ln 2 - \frac{1}{2}\ln\mathcal{N}\right) + \frac{1}{2}\ln\mathcal{P}(N) + \frac{1}{2}\ln\left(\frac{\det G_N^+}{\det G_N^-}\right),\tag{30}$$

where

$$\ln\mathcal{P}(N) \equiv \sum_{a=1}^{n-1}\ln\mathcal{P}_a(N) = \frac{1}{2}\sum_{a=1}^{n-1}\sum_{k=1}^{M_a}f(u_k^{(a)})\tag{31}$$

with $f(u) = \ln(\frac{u^2 + 1/4}{u^2})$. The first term in Eq. (30) admits the asymptotic expansion

$$\left[\frac{n}{2}\right]\ln 2 - \frac{1}{2}\ln\mathcal{N} = -\frac{N}{2}\ln n + \frac{n-1}{2}\ln 2 + \mathcal{O}(e^{-N/\xi}),\tag{32}$$

with $\xi = 1/\ln(\frac{n}{|n-4|})$ denoting the correlation length of $|\mathrm{MPS}\rangle$ [13], while the second and third terms explicitly depend on the Bethe roots of $|\psi_0\rangle$. However, in the large-$N$ limit, it is convenient to analyze the asymptotic behaviors of both the second and third terms using the method of nonlinear integral equations (NLIEs) [28, 33–38], which allows one to extract both the non-universal constant $\alpha$ and the universal subleading term $\ln g$ in Eq. (29).

The main idea of the NLIEs is to convert sums over Bethe roots of the form $\sum_k h(u_k^{(a)})$, where $h(u)$ is an analytic function, into contour integrals in the complex plane. For the ULS-chain ground state, the Bethe roots lie on the real axis and are symmetrically distributed about the origin. As a result, the associated counting functions $\phi_N^{(a)}(u)$ defined in Eq. (24) are analytic within the strip, $|\mathrm{Im}\,u| < \frac{1}{2}$, and satisfy the following analytic properties:

$$\phi_N^{(a)}(-u) = -\phi_N^{(a)}(u),\quad \left(\phi_N^{(a)}(u)\right)^* = \phi_N^{(a)}(u^*),\quad a = 1,\ldots,n-1.\tag{33}$$

The counting functions fully encode the Bethe roots via the BAEs [Eq. (23)], enabling the sum over Bethe roots to be expressed as a contour integral:

$$\sum_{k=1}^{M_a}h(u_k^{(a)}) = \frac{1}{2\pi i}\oint_{\mathscr{C}}h(z)\,\mathrm{d}\ln\left(1 + e^{iN\phi_N^{(a)}(z)}\right),\quad a = 1,\ldots,n-1,\tag{34}$$

where the integral contour $\mathscr{C}$ is chosen to encircle the entire real axis, as illustrated in Fig. 2(a), with $\xi \in (0, \frac{1}{2})$.

Applying the above contour integral trick [Eq. (34)] to the counting functions themselves, we arrive at

$$\phi_N^{(a)}(u) = \vartheta_1(u)\delta_{a,1} + \frac{1}{2\pi N i}\oint_{\mathscr{C}}\vartheta_1(u-z)\,\mathrm{d}\ln\left(1 + e^{iN\phi_N^{(a-1)}(z)}\right)$$

$$- \frac{1}{2\pi N i}\oint_{\mathscr{C}}\vartheta_2(u-z)\,\mathrm{d}\ln\left(1 + e^{iN\phi_N^{(a)}(z)}\right) + \frac{1}{2\pi N i}\oint_{\mathscr{C}}\vartheta_1(u-z)\,\mathrm{d}\ln\left(1 + e^{iN\phi_N^{(a+1)}(z)}\right),$$

$$\tag{35}$$

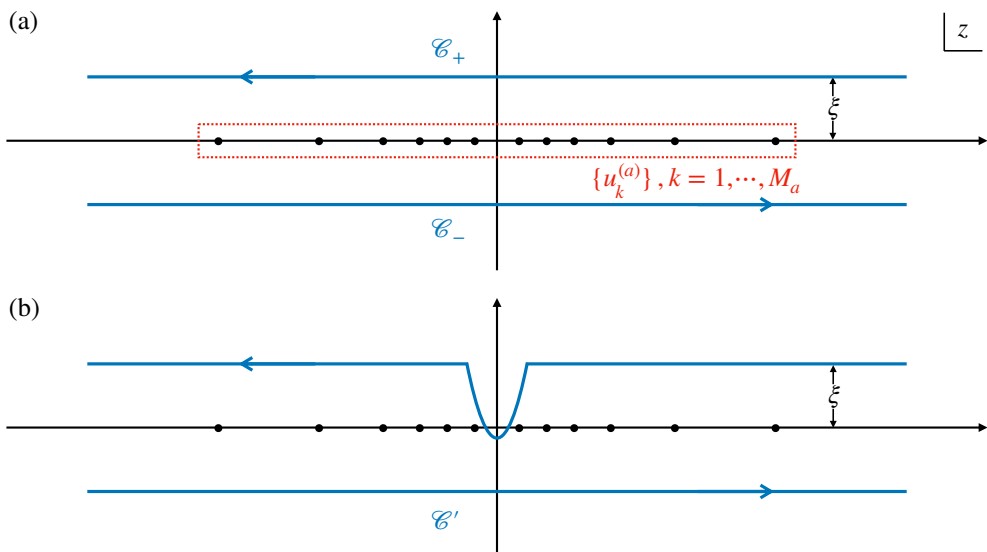

Figure 2: (a) The integral contour $\mathscr{C} = \mathscr{C}_+ + \mathscr{C}_-$ in Eq. (34), with $\xi \in (0, \frac{1}{2})$. (b) The deformed integral contour $\mathscr{C}'$ in Eq. (46), also with $\xi \in (0, \frac{1}{2})$.

which forms a set of integral equations imposing nonlinear constraints on the counting functions, thereby determining the distribution of Bethe roots at finite $N$.

Before proceeding with Eq. (35), we first examine its behavior in the thermodynamic limit. In this limit, we expect that the counting functions, as well as their derivatives converge to smooth functions. Specifically, in the thermodynamic limit, the auxiliary functions $\rho_N^{(a)}(u)$ [Eq. (27)] become Bethe root density distributions, $\rho^{(a)}(u) = \lim_{N \to \infty} \rho_N^{(a)}(u)$. Taking the derivative on both sides of Eq. (24) and using the continuous approximation

$$\frac{1}{N} = \frac{1}{2\pi}\left[\phi_N^{(a)}(u_{k+1}^{(a)}) - \phi_N^{(a)}(u_k^{(a)})\right] \approx \rho^{(a)}(u)\, du, \tag{36}$$

we arrive at a system of linear integral equations for the density functions:

$$\int_{-\infty}^{+\infty}\left[\delta_{ab}\delta + \frac{1}{2\pi}K_{ab}\right](u-v)\rho^{(b)}(v)dv = \frac{1}{2\pi}\frac{d}{du}\vartheta_1(u)\delta_{a,1}, \quad a = 1,\ldots,n-1, \tag{37}$$

where $K_{ab}(u)$ is defined in Eq. (27). Hereafter, we shall adopt the Einstein summation convention, where the summation over the dummy index [e.g., $b$ in Eq. (37)] is assumed. Convolving both sides of Eq. (37) with the inverse kernel

$$G_{ab}(u) = \left[\delta_{ab}\delta + \frac{1}{2\pi}K_{ab}\right]^{-1}(u) = \frac{1}{2\pi}\int_{-\infty}^{+\infty} dq\, e^{iqu}\tilde{G}_{ab}(q),$$

$$\tilde{G}_{ab}(q) = \frac{e^{|q|/2}}{n}\sum_{k=1}^{n-1}\frac{\sin(\pi ka/n)\sin(\pi kb/n)}{\cosh(q/2) - \cos(\pi k/n)}, \tag{38}$$

we obtain the Bethe root density distributions:

$$\rho^{(a)}(u) = \lim_{N \to \infty}\rho_N^{(a)}(u) = \frac{1}{n}\left[\frac{\sin(\pi a/n)}{\cosh(2\pi u/n) - \cos(\pi a/n)}\right], \quad a = 1,\ldots,n-1. \tag{39}$$

Integrating the densities yields the counting functions in the thermodynamic limit:

$$\phi^{(a)}(u) = \lim_{N \to \infty}\phi_N^{(a)}(u) = 2\arctan\left[\coth\left(\frac{\pi a}{2n}\right)\tanh\left(\frac{\pi u}{n}\right)\right], \quad a = 1,\ldots,n-1, \tag{40}$$

where we used the initial condition $\phi_N^{(a)}(0) = 0$.

Returning to Eq. (35), to separate out the contributions from the thermodynamic limit of the counting functions [Eq. (40)], we integrate by parts and convolve both sides with the kernel defined in Eq. (38). Using the analytic properties of the counting functions [Eq. (33)] and after some simplifications, we arrive at (see Appendix C)

$$\phi_N^{(a)}(u) = \phi^{(a)}(u) + \frac{2}{N}\operatorname{Im}\int_{-\infty}^{+\infty}\mathrm{d}v[\delta_{ab}\delta - G_{ab}](u - v - i\xi)\ln\left(1 + e^{iN\phi_N^{(b)}(v+i\xi)}\right), \quad u \in \mathbb{R},$$
(41)

with $\xi \in (0, \frac{1}{2})$, which is known as the NLIEs for the counting functions. It is obvious that the contribution from the nonlinear integral term on the right-hand-side of Eq. (41) vanishes as $N \to \infty$, yielding $\phi^{(a)}(u)$ in Eq. (40) as its solution in the thermodynamic limit. In fact, we are free to deform the integral contour by taking

$$\xi \to 0^+,$$
(42)

which allows for a more refined analysis of the nonlinear integral term in the thermodynamic limit:

$$\lim_{\xi \to 0^+}\lim_{N \to \infty}\operatorname{Im}\int_{-\infty}^{+\infty}\mathrm{d}v[\delta_{ab}\delta - G_{ab}](u - v - i\xi)\ln\left(1 + e^{iN\phi_N^{(b)}(v+i\xi)}\right),$$
(43)

where, the order of the limits, $\lim_{\xi \to 0^+}\lim_{N \to \infty}$, is important. Since we have

$$\phi^{(a)}(u + i\xi) = \phi^{(a)}(u) + 2\pi i\xi\rho^{(a)}(u) + \mathcal{O}(\xi^2)$$
(44)

with $\rho^{(a)}(u) > 0$, the integrand in Eq. (43) becomes exponentially suppressed, and thus the nonlinear term therein vanishes in this double limit, i.e., in the thermodynamic limit. This analysis can be extended to a general test function $h(u)$ that vanishes at infinity, $h(\pm\infty) = 0$. For such functions, we have

$$\lim_{\xi \to 0^+}\lim_{N \to \infty}\int_{-\infty}^{+\infty}h(u)\ln\left(1 + e^{iN\phi_N^{(a)}(u+i\xi)}\right)\mathrm{d}u = 0, \quad a = 1, \ldots, n-1.$$
(45)

A similar contour integral trick can be applied to the second contribution to the logarithm of the overlap [Eq. (31)]:

$$\ln\mathcal{P}_a(N) = \sum_{k=1}^{M_a}f(u_k^{(a)}) = \frac{1}{2\pi i}\oint_{\mathscr{C}'}f(z)\mathrm{d}\ln\left[1 + e^{iN\phi_N^{(a)}(z)}\right],$$
(46)

where the integral contour $\mathscr{C}'$ should be chosen to avoid the branch point of $f(u)$ at $u = 0$, as illustrated in Fig. 2(b). Integrating by parts and using the residue theorem, Eq. (46) can be expressed as an integral over the contour $\mathscr{C}$ instead [see Fig. 2(a)], together with the contribution from the residue:

$$\ln\mathcal{P}_a(N) = -\frac{1}{2\pi i}\oint_{\mathscr{C}}f'(z)\ln\left[1 + e^{iN\phi_N^{(a)}(z)}\right]\mathrm{d}z - \ln 2.$$
(47)

We can further split the contour $\mathscr{C}$ into upper and lower parts, $\mathscr{C}_+$ and $\mathscr{C}_-$. The integral over the lower contour can be rewritten as

$$-\frac{1}{2\pi i}\oint_{\mathscr{C}_-}f'(z)\ln\left[1 + e^{iN\phi_N^{(a)}(z)}\right]\mathrm{d}z$$

$$= -\frac{1}{2\pi i}\oint_{\mathscr{C}_-}f'(z)\ln\left[1 + e^{-iN\phi_N^{(a)}(z)}\right]\mathrm{d}z - \frac{N}{2\pi}\int_{-\infty}^{+\infty}f'(u - i\xi)\phi_N^{(a)}(u - i\xi)\mathrm{d}u.$$
(48)

Substituting the NLIEs [Eq. (41)] into the second integral at the right-hand-side of Eq. (48), we isolate the non-universal contribution to $\ln \mathcal{P}_a(N)$, which is proportional to $N$. Again, taking the double limit, $\lim_{\xi \to 0^+} \lim_{N \to \infty}$, we arrive at the asymptotic expansion:

$$\ln \mathcal{P}_a(N) = N \int_{-\infty}^{+\infty} f(u) \rho^{(a)}(u) \, du - \ln 2 + \cdots, \qquad (49)$$

where the ellipsis "$\cdots$" denotes the nonlinear integral terms, whose explicit forms are provided in Appendix C. These terms resemble Eq. (45) and vanish in the thermodynamic limit.

The third term in Eq. (30) is a ratio of factorized Gaudin determinants. Following the approach outlined in Ref. [37], we begin by rewriting it as

$$\frac{\det G_N^+}{\det G_N^-} = \frac{\det(\mathbb{I} - H_N^+)}{\det(\mathbb{I} - H_N^-)}, \qquad (50)$$

where the matrix indices run over the full set of Bethe roots, $\mathbb{I}$ denotes the identity matrix, and $H_N^{\pm}$ is defined as

$$H_N^{\pm} = \frac{P_N \pm Q_N}{2}, \qquad (51)$$

with

$$[P_N]_{ab;kl} \equiv P_{ab}(u_k^{(a)}, u_l^{(b)}) = -i \frac{\mathfrak{a}_N^{(a)}(u_k^{(a)})}{\mathfrak{a}_N^{(a)'}(u_k^{(a)})} K_{ab}(u_k^{(a)} - u_l^{(b)}),$$

$$[Q_N]_{ab;kl} \equiv Q_{ab}(u_k^{(a)}, u_l^{(b)}) = -i \frac{\mathfrak{a}_N^{(a)}(u_k^{(a)})}{\mathfrak{a}_N^{(a)'}(u_k^{(a)})} K_{ab}(u_k^{(a)} + u_l^{(b)}), \qquad (52)$$

where the auxiliary function $\mathfrak{a}_N^{(a)}(u)$ is defined as $\mathfrak{a}_N^{(a)}(u) = e^{iN\phi_N^{(a)}(u)}$, and $\mathfrak{a}_N^{(a)'}(u)$ denotes its derivative. Using the definition of $\phi_N^{(a)}(u)$ [Eq. (24)] and $K_{ab}(u)$ [Eq. (27)], one can show that $Q_N P_N = P_N Q_N$ and $Q_N Q_N = P_N P_N$. Therefore, the logarithm of the determinant ratio can be expanded as a series:

$$\ln \left[ \frac{\det G_N^+}{\det G_N^-} \right] = -\sum_{q=1}^{\infty} \frac{1}{q} \text{Tr}[(H_N^+)^q - (H_N^-)^q] = -\sum_{q=1}^{\infty} \text{Tr}[P_N^{q-1} Q_N], \qquad (53)$$

which admits the contour integral representation:

$$\text{Tr}[P_N^{q-1} Q_N]$$
$$= \sum_{a_1, \dots, a_q} \left[ \prod_{j=1}^q \frac{1}{2\pi i} \oint_{\mathscr{C}} d \ln \left( 1 + \mathfrak{a}_N^{(a_j)}(z_j) \right) \right] P_{a_1 a_2}(z_1, z_2) \cdots P_{a_{q-1} a_q}(z_{q-1}, z_q) Q_{a_q a_1}(z_q, z_1)$$
$$= \sum_{a_1, \dots, a_q} \left[ \prod_{j=1}^q \oint_{\mathscr{C}} \frac{dz_j}{2\pi} \frac{-\mathfrak{a}_N^{(a_j)'}(z_j)}{1 + \mathfrak{a}_N^{(a_j)}(z_j)} \right] K_{a_1 a_2}(z_1 - z_2) \cdots K_{a_{q-1} a_q}(z_{q-1} - z_q) K_{a_q a_1}(z_n + z_1). \qquad (54)$$

Taking the double limit, we have

$$\lim_{\xi \to 0^+} \lim_{N \to \infty} \frac{\mathfrak{a}_N^{(a)}(u + i\xi)}{1 + \mathfrak{a}_N^{(a)}(u + i\xi)} = 0, \quad \lim_{\xi \to 0^+} \lim_{N \to \infty} \frac{\mathfrak{a}_N^{(a)}(u - i\xi)}{1 + \mathfrak{a}_N^{(a)}(u - i\xi)} = 1, \quad u \in \mathbb{R}, \quad a = 1, 2, \qquad (55)$$

so that only the integration along the lower contour contributes:

$$
\lim_{N\to\infty} \mathrm{Tr}[P_N^{q-1} Q_N]
$$

$$
= (-1)^q \sum_{a_1,\ldots,a_q} \left[ \prod_{j=1}^{q} \int_{-\infty}^{+\infty} \frac{\mathrm{d}u_j}{2\pi} \right] K_{a_1 a_2}(u_1 - u_2) \cdots K_{a_{q-1} a_q}(u_{q-1} - u_q) K_{a_q a_1}(u_q + u_1)
$$

$$
= \frac{(-1)^q}{2} \mathrm{Tr}\left[ \left( \int_{-\infty}^{+\infty} \frac{\mathrm{d}u}{2\pi} K(u) \right)^q \right], \tag{56}
$$

where, in the second equality, we introduced a change of variables: $u_1' = u_1 - u_2, \ldots,$ $u_{q-1}' = u_{q-1} - u_q$, $u_q' = u_1 + u_q$. The kernel integral evaluates to

$$
\int_{-\infty}^{+\infty} \frac{\mathrm{d}u}{2\pi} K_{ab}(u) = \delta_{ab} - (\delta_{a+1,b} + \delta_{a-1,b}), \quad a, b = 1, \ldots, n-1, \tag{57}
$$

which is a tridiagonal matrix with eigenvalues $\lambda_a = 1 - 2\cos(\pi a/n)$. This leads to

$$
\lim_{N\to\infty} \ln\left[ \frac{\det G_N^+}{\det G_N^-} \right] = \frac{1}{2} \sum_{a=1}^{n-1} \ln(1 + \lambda_a) = \ln\left[ \prod_{a=1}^{n-1} 2\sin\left( \frac{\pi a}{2n} \right) \right] = \ln\sqrt{n}. \tag{58}
$$

Combining the results in Eqs. (32), (49), and (58), we obtain the asymptotic expansion of the logarithm of the overlap in Eq. (29), with the non-universal surface free energy density

$$
\alpha = \frac{1}{2}\ln n + \sum_{a=1}^{n-1} \frac{\sin(\pi a/n)}{2n} \int_0^\infty \mathrm{d}u \frac{\ln[u^2/(u^2 + 1/4)]}{\cosh(2\pi u/n) - \cos(\pi a/n)}
$$

$$
= \frac{1}{2}\ln\left[ n \prod_{a=1}^{n-1} \frac{(2\pi)^{1-a/n}\Gamma\left(1 - \frac{a}{2n}\right)}{\Gamma\left(\frac{a}{2n}\right)} \right] - \sum_{a=1}^{n-1} \frac{\sin(\pi a/n)}{4\pi} \int_0^\infty \mathrm{d}u \frac{\ln\left[u^2 + (\pi/n)^2\right]}{\cosh u - \cos(\pi a/n)}, \tag{59}
$$

and the universal subleading term

$$
\ln g = \frac{1}{4}\ln n, \tag{60}
$$

which yields the expected Affleck-Ludwig entropy associated with the non-Cardy boundary states $|\mathcal{D}_\sigma\rangle$ ($|\mathcal{D}_{\sigma_\pm}\rangle$) for odd (even) $n$, as given in Eqs. (9) and (16), respectively.

Although we have extracted the Affleck-Ludwig entropy directly in the large-$N$ limit by fully exploiting integrability, it is also of interest to examine the finite-size corrections on top of Eq. (29). Such finite-size corrections are expected to exhibit universal scaling behavior in the regime $N \gg 1$. To investigate this, we consider the subtracted logarithmic overlap, $\ln|\langle\psi_0|\mathrm{MPS}\rangle| + \alpha N$, where the overlap formula [Eq. (25)] enables us to compute this quantity with high precision for large system sizes by numerically solving the BAEs in Eq. (23). The results show a linear dependence on $1/\ln N$ for large $N$, as illustrated in Fig. 2. This is a typical logarithmic correction arising from marginally irrelevant perturbations due to lattice effects in the ULS chain [12, 39]. A linear fit to the data for $n = 3, 4, 5, 6$ in the range $N \in [800, 1200]$ yields a relative error of approximately 1% compared to the exact results in the thermodynamic limit.

## 5   Summary and outlook

To summarize, we have constructed a class of non-Cardy conformal boundary states in $\mathrm{SU}(n)_1$ WZW CFT via the symmetry embedding $\mathrm{Spin}(n)_2 \subset \mathrm{SU}(n)_1$. We have also identified their

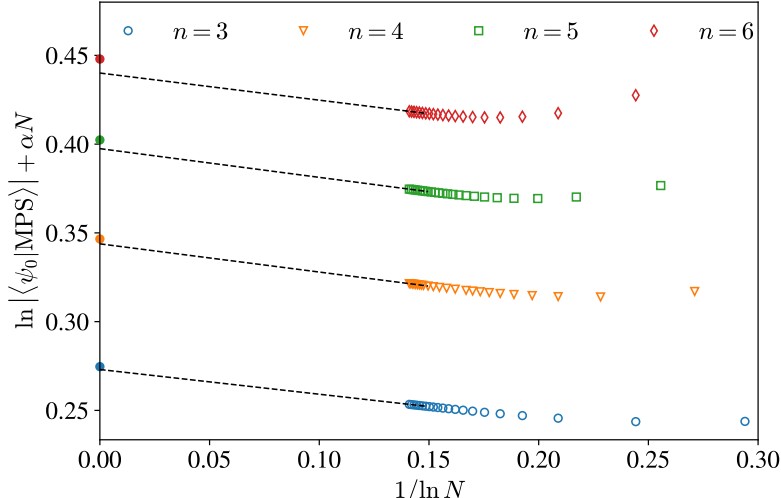

Figure 3: The subtracted logarithmic overlap, $\ln|\langle\psi_0|\text{MPS}\rangle| + \alpha N$, is plotted against $1/\ln N$ for $n = 3, 4, 5, 6$. The black dashed lines represent linear fits to the data in the range $N \in [800, 1200]$. For reference, the exact large-$N$ values, $\ln g = \frac{1}{4}\ln n$, are indicated by solid dots on the vertical axis.

lattice realizations as SO($n$) AKLT states for the integrable SU($n$) ULS chains. Using integrability techniques, we have analytically computed the Affleck-Ludwig boundary entropy through wave function overlaps, finding perfect agreement with CFT predictions. Our results establish a concrete link between symmetry embedding, integrable models, and boundary CFT, and offer new input toward the full classification of conformal boundary states.

These results are inspired by Affleck's pioneering insights into boundary CFT and AKLT states, which fundamentally shaped our understanding of critical and gapped phases in one-dimensional systems.

For the non-Cardy boundary states arising from the embedding Spin($n$)$_2 \subset$ SU($n$)$_1$, it would be interesting to investigate whether the SO($n$) AKLT states also serve as integrable boundary states for the SU($n$) Haldane-Shastry model [40–43], potentially allowing analytic computation of the wave function overlaps and the Affleck-Ludwig entropy. More broadly, constructions similar to those presented in this work and in Ref. [8] may be applicable to other conformal embeddings or even to higher-dimensional systems.

## Acknowledgments

H.-H.T. acknowledges helpful discussions with Philippe Lecheminant, Masaki Oshikawa, Thomas Quella, and Germán Sierra. Y.S.Z. is supported by the Sino-German (CSC-DAAD) Postdoc Scholarship Program. Y.H.W. is supported by NNSF of China under Grant No. 12174130.

## A    Fusion rules from embedding Spin($n$)$_2 \subset$ SU($n$)$_1$

In this Appendix, we derive the fusion rules of TDLs in Eqs. (4) and (13) using the conformal embedding Spin($n$)$_2 \subset$ SU($n$)$_1$.

We begin with a brief review of the fusion algebras of SU($n$)$_1$ and Spin($n$)$_2$, whose chiral primary fields and their conformal weights have been introduced in Sec. 2. The fusion rules

of $SU(n)_1$ take a simple form:

$$[a] \times [b] = [(a+b) \bmod n], \quad a, b = 0, \ldots, n-1. \tag{A.1}$$

The fusion rules of $Spin(n)_2$ differ between the odd $n \equiv 2r + 1$ and even $n \equiv 2r + 2$ cases. However, in both cases, the chiral primary field $Z$ is a simple current, which fuses with itself to yield the identity and leaves each $Y_m$ invariant:

$$Z \times Z = I, \quad Z \times Y_m = Y_m, \quad m = 1, \ldots, r. \tag{A.2}$$

For odd $n$, the twisted fields $X$ and $X'$ are both self-conjugate and transform to each other under the action of the simple current:

$$Z \times X = X', \quad X \times X = I + \sum_{m=1}^{r} Y_m. \tag{A.3}$$

For even $n$, the simple current leaves $W_\pm$ invariant and exchanges the twisted fields $X_\pm$ and $X'_\pm$:

$$Z \times W_\pm = W_\pm, \quad Z \times X_\pm = X'_\pm. \tag{A.4}$$

The fields $X_\pm$ and $X'_\pm$ are self-conjugate for odd $r$, while conjugate to each other for even $r$. The corresponding fusion rules read

$$X_\pm \times X_\pm = I + \sum_{m=1 \,(m\,\text{even})}^{r} Y_m + W_\pm, \quad \text{odd } r,$$

$$X_+ \times X_- = I + \sum_{m=1 \,(m\,\text{even})}^{r} Y_m, \quad \text{even } r. \tag{A.5}$$

We analyze the fusion rules of TDLs in the tree channel, where the CFT is quantized on a circle and the TDLs are operators extended along the spatial direction. In the operator formalism, the fusion of TDLs corresponds to their operator product, e.g., $D_\sigma \times D_\sigma \equiv D_\sigma D_\sigma$, and the charge conjugate of a TDL corresponds to its Hermitian conjugate. In our case, we have $\mathcal{D}_\sigma^\dagger = \mathcal{D}_\sigma$ for odd $n$, and $\mathcal{D}_{\sigma_\pm}^\dagger = \mathcal{D}_{\sigma_\pm}$ for even $n$ with odd $r$, while $\mathcal{D}_{\sigma_+}^\dagger = \mathcal{D}_{\sigma_-}$ for even $n$ with even $r$.

For odd $n$, the fusion of the TDL, $\mathcal{D}_\sigma = \frac{1}{2}(\mathcal{D}_X + \mathcal{D}_{X'})$, with itself reads

$$\mathcal{D}_\sigma \mathcal{D}_\sigma = \frac{1}{4}(\mathcal{D}_X \mathcal{D}_X + \mathcal{D}_{X'} \mathcal{D}_{X'} + 2\mathcal{D}_X \mathcal{D}_{X'}). \tag{A.6}$$

Here, $\mathcal{D}_X$ and $\mathcal{D}_{X'}$ are Verlinde lines in $Spin(2r+1)_2$, and thus their fusion rules coincide with those of the corresponding chiral primary fields, i.e., Eq. (A.3). Explicitly, we have

$$\mathcal{D}_X \mathcal{D}_X = \mathcal{D}_{X'} \mathcal{D}_{X'} = \mathcal{D}_I + \sum_{m=1}^{r} \mathcal{D}_{Y_m},$$

$$\mathcal{D}_X \mathcal{D}_{X'} = \mathcal{D}_Z + \sum_{m=1}^{r} \mathcal{D}_{Y_m}. \tag{A.7}$$

Meanwhile, under the conformal embedding $Spin(2r+1)_2 \subset SU(2r+1)_1$, the Verlinde lines in the two theories are related via the identifications:

$$\mathcal{D}_{[0]} = \frac{1}{2}(\mathcal{D}_I + \mathcal{D}_Z), \quad \mathcal{D}_{[m]} + \mathcal{D}_{[2r+1-m]} = \mathcal{D}_{Y_m}, \quad m = 1, \ldots, r. \tag{A.8}$$

Substituting Eq. (A.7) and (A.8) into Eq. (A.6), we arrive at

$$\mathcal{D}_\sigma \mathcal{D}_\sigma = \sum_{a=0}^{2r} \mathcal{D}_{[a]}, \tag{A.9}$$

which completes the proof of Eq. (4). We also note that the normalization factor "$\frac{1}{2}$" of $\mathcal{D}_\sigma$ is chosen to ensure that it is a simple TDL in $\mathrm{SU}(n)_1$, i.e., the fusion multiplicity of the identity line $\mathcal{D}_{[0]}$ equals one.

For even $n$, the embedding decompositions of Verlinde lines in the two theories read

$$\mathcal{D}_{[0]} = \frac{1}{2}(\mathcal{D}_I + \mathcal{D}_Z), \quad \mathcal{D}_{[r+1]} = \frac{1}{2}(\mathcal{D}_{W_+} + \mathcal{D}_{W_-}), \quad \mathcal{D}_{[m]} + \mathcal{D}_{[2r+2-m]} = \mathcal{D}_{Y_m}, \quad m = 1,\ldots,r. \tag{A.10}$$

Following an analysis analogous to the odd $n$ case, we obtain the fusion rules for the simple TDLs in this setting:

$$\mathcal{D}_{\sigma_\pm} \mathcal{D}_{\sigma_\pm} = \sum_{\substack{a=0\,(a\,\mathrm{even})}}^{2r+1} \mathcal{D}_{[a]}, \quad \mathrm{odd}\ r,$$

$$\mathcal{D}_{\sigma_+} \mathcal{D}_{\sigma_-} = \sum_{\substack{a=0\,(a\,\mathrm{even})}}^{2r+1} \mathcal{D}_{[a]}, \quad \mathrm{even}\ r, \tag{A.11}$$

which are precisely the results in Eq. (13).

Any $\mathrm{SU}(n)_1$ Cardy state can be generated by acting on the identity Cardy state with the corresponding Verlinde line: $|[a]\rangle = \mathcal{D}_{[a]}|[0]\rangle$, whose overlap with $|[0]\rangle$ yields the corresponding chiral character in the loop channel: $\langle[0]|e^{-\beta H}|[a]\rangle = \chi_{[a]}(q)$ with $q = e^{-\pi L/\beta}$. Combining this with the fusion rules in Eqs. (A.9) and (A.11), we arrive at the BCFT partition functions $\mathcal{Z}_{\mathcal{D}_\sigma,\mathcal{D}_\sigma}$ [Eq. (8)] and $\mathcal{Z}_{\mathcal{D}_{\sigma_\pm},\mathcal{D}_{\sigma_\pm}}$ [Eq. (15)], respectively.

The conformal embedding also implies that the $\mathrm{SU}(n)_1$ identity Cardy state decomposes as a linear combination of $\mathrm{Spin}(n)_2$ Cardy states: $|[0]\rangle = \frac{1}{\sqrt{2}}(|I\rangle + |Z\rangle)$. Together with the $\mathrm{Spin}(n)_2$ fusion rules, $Z \times X = X'$ (odd $n$) and $Z \times X_\pm = X'_\pm$ (even $n$), this leads to the embedding decomposition of $|\mathcal{D}_\sigma\rangle$ [Eq. (5)] and $|\mathcal{D}_{\sigma_\pm}\rangle$ [Eq. (14)].

# B  Integrability and overlap formula

In this Appendix, we briefly review the integrability of the $\mathrm{SU}(n)$ ULS chain and the overlap formula for the $\mathrm{SO}(n)$ AKLT states as integrable boundary states.

For the $\mathrm{SU}(n)$ ULS chain, the $R$-matrix is defined by

$$R_{i,j}(\lambda) = \lambda \, \mathbb{1}_{i,j} + P_{i,j}, \tag{B.1}$$

where $\lambda$ is the spectral parameter, $\mathbb{1}$ is the identity operator, and $P_{i,j}$ is the permutation operator acting on two $\mathrm{SU}(n)$ spins at site $i$ and $j$, defined by $P_{i,j}|a\rangle_i \otimes |b\rangle_j = |b\rangle_j \otimes |a\rangle_i$ with $a,b = 0,\ldots,n-1$. Accordingly, the transfer matrix is defined as

$$T(\lambda) = \mathrm{Tr}_0\left[R_{0,1}(\lambda)\cdots R_{0,N}(\lambda)\right], \tag{B.2}$$

where the trace is taken over the auxiliary $\mathrm{SU}(n)$ spin space. Since transfer matrices commute for all spectral parameters, i.e., $[T(\lambda), T(\mu)] = 0 \ \forall \lambda, \mu$, the expansion of $T(\lambda)$ in powers of $\lambda$ generates a family of mutually commuting operators, among which the ULS-chain Hamiltonian is included.

An integrable boundary state $|\varphi\rangle$ of a given integrable lattice model is defined by [24]

$$T(\lambda)|\varphi\rangle = \Pi T(\lambda)\Pi |\varphi\rangle, \tag{B.3}$$

where $T(\lambda)$ is the transfer matrix and $\Pi$ is the spatial reflection operator.

For the ULS chain, we express the $R$-matrix as $R(\lambda) = \sum_{a,b=0}^{n-1} e_{ab} \otimes R_{ab}(\lambda)$, where $e_{ab}$ denotes the $n \times n$ matrix with a single nonzero entry $(e_{ab})_{ab} = 1$ and zeros elsewhere, and the indices $a, b$ outside the parentheses indicate matrix elements. Both $T(\lambda)$ and $\Pi T(\lambda)\Pi$ can be viewed as matrix product operators, with local tensors $R_{ab}$ and $R_{ab}^t$, respectively, where $R_{ab}^t$ denotes the transpose of $R_{ab}$. The state $|\text{MPS}\rangle$ defined in Eq. (19) has been proven to be an integrable boundary state of the ULS chain, and its integrability condition Eq. (B.3) can be equivalently formulated in terms of the local tensor as follows [26]:

$$[R_{ab}(\lambda)\Gamma^b]\mathbf{K}(\lambda) = \mathbf{K}(\lambda)[R_{ab}^t(\lambda)\Gamma^b], \tag{B.4}$$

which is known as the $KT$-relation. Here, the $K$-matrix, $\mathbf{K}(\lambda)$, is expressed as

$$\mathbf{K}(\lambda) = \sum_{a,b=0}^{n-1} e_{ab} \otimes \mathbf{K}_{ab}(\lambda), \quad \mathbf{K}_{ab}(\lambda) = \Gamma^a\Gamma^b + 2\lambda\delta_{ab}\mathbb{I}, \tag{B.5}$$

where the indices $a, b$ now label the auxiliary space of the $R$-matrix, and $\mathbb{I}$ is the identity matrix. For even-$n$ cases, the $K$-matrix commutes with the additional Gamma matrix,

$$[\mathbf{K}_{ab}(\lambda), \Gamma^n] = 0, \tag{B.6}$$

so the state $|\widetilde{\text{MPS}}\rangle$ defined in Eq. (19) also satisfies the integrability condition [Eq. (B.3)]. This proves that the AKLT states defined in Eq. (18) are integrable boundary states of the SU($n$) ULS chain for both odd and even $n$.

The overlap formula for an integrable boundary state can be derived algebraically once the $KT$-relation is established [32, 44, 45]. In fact, the $K$-matrix [Eq. (B.5)] satisfies the twisted boundary Yang-Baxter equations and thereby forms a representation of the twisted Yangian algebra, $Y(\mathfrak{su}(n), \mathfrak{so}(n))$, where a set of $F$-operators, $\{\mathbf{F}^{(a)}(\lambda)\}$, $a = 1, \ldots, n-1$, generates the commuting subalgebra [26, 31, 32]. In the algebraic Bethe ansatz for the ULS chain, the pseudo-vacuum is chosen as

$$|\Omega\rangle \equiv \prod_{j=1}^{N} |0\rangle_j. \tag{B.7}$$

Its overlap with $|\text{MPS}\rangle$ [Eq. (19)] is given by $\langle\Omega|\text{MPS}\rangle = \frac{1}{\sqrt{\mathcal{N}}}\text{Tr}(\mathbf{B}^N)$, where $\mathbf{B} = \Gamma^0$ is the pseudo-vacuum overlap operator that commutes with all $F$-operators.

Only the Bethe eigenstates $|\{u\}\rangle$ with zero lattice momentum can have a non-vanishing overlap with $|\text{MPS}\rangle$. These Bethe eigenstates have paired Bethe roots, $\{u_k^{(a)}\} = \{u_{k_+}^{(a)}\}\cup\{-u_{k_+}^{(a)}\}$. The overlap formula between $|\{u\}\rangle$ and $|\text{MPS}\rangle$ takes the form [32, 46]:

$$\langle\{u\}|\text{MPS}\rangle = \frac{1}{\sqrt{\mathcal{N}}}\left[\sum_{l=1}^{\chi} \beta_l^N \mathcal{F}_l(\{u_+\})\right]\sqrt{\frac{\det G_N^+}{\det G_N^-}}, \tag{B.8}$$

where $\chi = 2^{\left[\frac{n}{2}\right]}$ is the MPS bond dimension, and $G_N^{\pm}(\{u_+\})$ are factorized Gaudin matrices defined in Eq. (26). The extra prefactor depends on the specific structure of the integrable MPS. The coefficients $\beta_l = \pm 1$ ($l = 1, \ldots, \chi$) are eigenvalues of the pseudo-vacuum overlap operator $\mathbf{B}$. As we have chosen $\text{mod}(N, 2n) = 0$, these eigenvalues satisfy

$$\beta_l^N = 1, \quad l = 1, \ldots, \chi. \tag{B.9}$$

The factor $\mathcal{F}_l(\{u_+\})$ is given by

$$\mathcal{F}_l(\{u_+\}) = \prod_{a=1}^{n-1} \prod_{k_+=1}^{M_a/2} \left[ f_l^{(a)}\left(-iu_{k_+}^{(a)} - \frac{a}{2}\right) \sqrt{\frac{(u_{k_+}^{(a)})^2}{(u_{k_+}^{(a)})^2 + 1/4}} \right], \quad l = 1, \dots, \chi, \qquad (B.10)$$

where $f_l^{(a)}(\lambda)$ denotes the $l$-th eigenvalue of the corresponding $F$-operator $\mathbf{F}^{(a)}(\lambda)$.

The $F$-operators can be constructed from the $K$-matrix. To this end, one can introduce a set of nested $K$-matrices $\mathbf{K}_{ab}^{(k)}(\lambda)$, with $k = 0, \dots, n-1$, defined recursively by

$$\mathbf{K}_{ab}^{(k+1)}(\lambda) = \mathbf{K}_{ab}^{(k)}(\lambda) - \mathbf{K}_{ak}^{(k)}(\lambda)[\mathbf{K}_{kk}^{(0)}(\lambda)]^{-1}\mathbf{K}_{kb}^{(k)}(\lambda) \qquad (B.11)$$

with the initial condition $\mathbf{K}_{ab}^{(0)}(\lambda) \equiv \mathbf{K}_{ab}(\lambda)$. The $F$-operators are then defined as

$$\mathbf{F}^{(a)}(\lambda) = [\mathbf{K}_{a-1,a-1}^{(a-1)}(\lambda)]^{-1}\mathbf{K}_{aa}^{(a)}(\lambda), \quad a = 1, \dots, n-1. \qquad (B.12)$$

Solving the recurrence relation reveals that all $F$-operators are proportional to the identity, with eigenvalues

$$f_l^{(a)}(\lambda) = \frac{(2\lambda + a - 1)(2\lambda + a + 1)}{(2\lambda + a)^2}, \quad l = 1, \dots, \chi, \quad a = 1, \dots, n-1. \qquad (B.13)$$

Substituting Eq. (B.13) into Eq. (B.10), we arrive at

$$\mathcal{F}_l(\{u_+\}) = \prod_{a=1}^{n-1} \prod_{k_+=1}^{M_a/2} \sqrt{\frac{(u_{k_+}^{(a)})^2 + 1/4}{(u_{k_+}^{(a)})^2}}, \quad l = 1, \dots, \chi. \qquad (B.14)$$

Finally, substituting Eq. (B.14) into Eq. (B.8) yields the overlap formula as presented in Eq. (25).

## C   Detailed derivations of the nonlinear integral equations

In this Appendix, we provide detailed derivations of the NLIEs for the counting functions [Eq. (41)], as well as the contour integral representation of $\ln\mathcal{P}_a$ [Eq. (49)].

To derive Eq. (41), we begin with the contour integral representation [Eq. (35)]. We perform integration by parts for each contour integral in Eq. (35) and decompose them into three contributions:

$$\frac{1}{2\pi Ni} \oint_{\mathscr{C}} \vartheta_p(u-z) \mathrm{d}\ln\left(1 + e^{iN\phi_N^{(a)}(z)}\right)$$

$$= \frac{1}{2\pi Ni} \oint_{\mathscr{C}} \vartheta_p'(u-z) \ln\left(1 + e^{iN\phi_N^{(a)}(z)}\right) \mathrm{d}z$$

$$= \frac{1}{2\pi Ni} \oint_{\mathscr{C}_+} \vartheta_p'(u-z) \ln\left(1 + e^{iN\phi_N^{(a)}(z)}\right) \mathrm{d}z$$

$$+ \frac{1}{2\pi Ni} \oint_{\mathscr{C}_-} \vartheta_p'(u-z) \ln\left(1 + e^{-iN\phi_N^{(a)}(z)}\right) \mathrm{d}z + \frac{1}{2\pi} \int_{-\infty}^{+\infty} \vartheta_p'(u-v)\phi_N^{(a)}(v)\mathrm{d}v. \qquad (C.1)$$

Substituting Eq. (C.1) into Eq. (35), we arrive at

$$
\int_{-\infty}^{+\infty} \left[ \delta_{ab}\delta + \frac{1}{2\pi}K_{ab} \right](u-v)\phi_N^{(b)}(v)\mathrm{d}v
$$
$$
= \vartheta_1(u)\delta_{a,1}
$$
$$
- \frac{1}{2\pi i}\left[ \oint_{\mathscr{C}_+} K_{ab}(u-z)\ln\left(1 + e^{iN\phi_N^{(b)}(z)}\right)\mathrm{d}z + \oint_{\mathscr{C}_-} K_{ab}(u-z)\ln\left(1 + e^{-iN\phi_N^{(b)}(z)}\right)\mathrm{d}z \right],
$$
(C.2)

where the auxiliary function $K_{ab}(u)$ has been defined in Eq. (27). The NLIEs are derived by acting with the convolution kernel $G_{ab}(u)$ [Eq. (38)] on both sides of Eq. (C.2):

$$
\phi_N^{(a)}(u) = \phi^{(a)}(u) - \frac{i}{N}\int_{-\infty}^{+\infty}\mathrm{d}v[\delta_{ab}\delta - G_{ab}](u-v-i\xi)\ln\left(1 + e^{iN\phi_N^{(b)}(v+i\xi)}\right)
$$
$$
+ \frac{i}{N}\int_{-\infty}^{+\infty}\mathrm{d}v[\delta_{ab}\delta - G_{ab}](u-v+i\xi)\ln\left(1 + e^{-iN\phi_N^{(b)}(v-i\xi)}\right),
$$
(C.3)

with $\xi \in (0, \frac{1}{2})$, where we used

$$
\oint_{\mathscr{C}_\pm} h(z)\mathrm{d}z = \mp \int_{-\infty}^{+\infty} h(u \pm i\xi)\mathrm{d}u,
$$
(C.4)

and

$$
\frac{1}{2\pi}\int_{-\infty}^{+\infty}\mathrm{d}v\, G_{ab}(u-v)K_{bc}(v-w) = \int_{-\infty}^{+\infty}\mathrm{d}v\, G_{ab}(u-v)\left[(G^{-1})_{bc} - \delta_{bc}\delta\right](v-w)
$$
$$
= [\delta_{ac}\delta - G_{ac}](u-w).
$$
(C.5)

Specifically, for real values of $u$, using the analytic properties of the counting functions [Eq. (33)], Eq. (C.3) can be rewritten as

$$
\phi_N^{(a)}(u) = \phi^{(a)}(u) + \frac{2}{N}\,\mathrm{Im}\int_{-\infty}^{+\infty}\mathrm{d}v[\delta_{ab}\delta - G_{ab}](u-v-i\xi)\ln\left(1 + e^{iN\phi_N^{(b)}(v+i\xi)}\right), \quad u \in \mathbb{R},
$$
(C.6)

which yields the NLIEs for counting functions as presented in Eq. (41).

To derive Eq. (49), we first rewrite the contour integral in Eq. (47) as

$$
-\frac{1}{2\pi i}\oint_{\mathscr{C}} f'(z)\ln\left[1 + e^{iN\phi_N^{(a)}(z)}\right]\mathrm{d}z
$$
$$
= \frac{1}{\pi}\,\mathrm{Im}\int_{-\infty}^{+\infty}\mathrm{d}u\, f'(u+i\xi)\ln\left[1 + e^{iN\phi_N^{(a)}(u+i\xi)}\right] - \frac{N}{2\pi}\oint_{\mathscr{C}_-} f'(z)\phi_N^{(a)}(z)\mathrm{d}z,
$$
(C.7)

with $0 < \xi < \frac{1}{2}$. Substituting the NLIEs for counting functions [Eq. (C.3)] into the second term

at the right-hand-side of Eq. (C.7), we obtain

$$
-\frac{N}{2\pi}\oint_{\mathscr{C}_-} f'(z)\phi_N^{(a)}(z)\mathrm{d}z
$$

$$
=-\frac{N}{2\pi}\oint_{\mathscr{C}_-} f'(z)\phi^{(a)}(z)\mathrm{d}z
$$

$$
-\frac{1}{2\pi i}\oint_{\mathscr{C}_-} \mathrm{d}z f'(z)\int_{-\infty}^{+\infty} \mathrm{d}v[\delta_{ab}\delta-G_{ab}](z-v-i\xi)\ln\left[1+e^{iN\phi_N^{(b)}(v+i\xi)}\right]
$$

$$
+\frac{1}{2\pi i}\oint_{\mathscr{C}_-} \mathrm{d}z f'(z)\int_{-\infty}^{+\infty} \mathrm{d}v[\delta_{ab}\delta-G_{ab}](z-v+i\xi)\ln\left[1+e^{-iN\phi_N^{(b)}(v-i\xi)}\right]. \qquad (C.8)
$$

For the first term in Eq. (C.8), we shift the contour $\mathscr{C}_-$ to the real axis, noting that $\phi^{(a)}(0)=0$, and then integrate by parts

$$
-\frac{N}{2\pi}\oint_{\mathscr{C}_-} f'(z)\phi^{(a)}(z)\mathrm{d}z = -\frac{N}{2\pi}\int_{-\infty}^{+\infty} f'(u)\phi^{(a)}(u)\mathrm{d}u = N\int_{-\infty}^{+\infty} f(u)\rho^{(a)}(u)\mathrm{d}u. \qquad (C.9)
$$

For the second term in Eq. (C.8), we use the residue theorem to rewrite the integral over contour $\mathscr{C}_-$ to one over $\mathscr{C}_+$, picking up the residue at $z=0$:

$$
-\frac{1}{2\pi i}\oint_{\mathscr{C}_-} f'(z)[\delta_{ab}\delta-G_{ab}](z-v-i\xi)\mathrm{d}z
$$

$$
=[\delta_{ab}\delta-G_{ab}](-v-i\xi)+\frac{1}{2\pi i}\oint_{\mathscr{C}_+} f'(z)[\delta_{ab}\delta-G_{ab}](z-v-i\xi)\mathrm{d}z. \qquad (C.10)
$$

Substituting Eqs. (C.9) and (C.10) into Eq. (C.8), we arrive at

$$
-\frac{N}{2\pi}\oint_{\mathscr{C}_-} f'(z)\phi_N^{(a)}(z)\mathrm{d}z
$$

$$
= N\int_{-\infty}^{+\infty} f(u)\rho^{(a)}(u)\mathrm{d}u + 2\,\mathrm{Re}\int_0^{+\infty} \mathrm{d}u[\delta_{ab}\delta-G_{ab}](-u-i\xi)\ln\left[1+e^{iN\phi_N^{(b)}(u+i\xi)}\right]
$$

$$
-\frac{1}{\pi}\,\mathrm{Im}\int_{-\infty}^{+\infty}\mathrm{d}u f'(u+i\xi)\int_{-\infty}^{+\infty}\mathrm{d}v[\delta_{ab}\delta-G_{ab}](u-v)\ln\left[1+e^{iN\phi_N^{(b)}(v+i\xi)}\right], \qquad (C.11)
$$

where we used the analytic properties of the counting functions [Eq. (33)].

Substituting Eq. (C.11) back into Eq. (C.7) yields the contour integral representation of $\ln\mathcal{P}_a(N)$, as given in Eq. (49):

$$
\ln\mathcal{P}_a(N) = N\int_{-\infty}^{+\infty} f(u)\rho^{(a)}(u)\,\mathrm{d}u - \ln 2
$$

$$
+2\,\mathrm{Re}\int_0^{+\infty} \mathrm{d}u\,[\delta_{ab}\delta-G_{ab}](-u-i\xi)\ln\left[1+e^{iN\phi_N^{(b)}(u+i\xi)}\right]
$$

$$
+\frac{1}{\pi}\,\mathrm{Im}\int_{-\infty}^{+\infty}\mathrm{d}u\,f'(u+i\xi)\int_{-\infty}^{+\infty}\mathrm{d}v\,G_{ab}(u-v)\ln\left[1+e^{iN\phi_N^{(b)}(v+i\xi)}\right], \qquad (C.12)
$$

where the last two nonlinear integral terms vanish in the thermodynamic limit.

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
