# Peer review of "SO(n) Affleck-Kennedy-Lieb-Tasaki states as conformal boundary states of integrable SU(n) spin chains"

_SciPost Physics_

## Round 2 · Referee Report · Anonymous (Referee 1) · 2025-11-23

Report

The paper studies boundary states in WZW models, their properties and their links to spin chains, in particular AKLT states, thus providing a synergetic link between different research areas. The presentation is good and, as far as I can tell, the results are correct. In particular, the manuscript contains an informative introduction as well as conclusion, which indicates a clear path for follow-up works.

Recommendation

Publish (meets expectations and criteria for this Journal)

---

## Round 2 · Referee Report · Anonymous (Referee 2) · 2025-12-1

Disclosure of Generative AI use

The referee discloses that the following generative AI tools have been used in the preparation of this report:

I used ChatGPT (GPT-5 mini) to help rephrase certain sentences for clarity and grammar in the referee report. The AI was used only for language refinement, and all scientific evaluation and content were provided solely by the reviewer.

Strengths

  1. A novel construction of non-Cardy conformal boundary states in the SU$(n)_1$ WZW theory via the conformal embedding Spin$(n)_2$ $\subset$ SU$(n)_1$.

  2. Extraction of the Affleck-Ludwig boundary entropy by computing overlaps between the SU$(n)$ Uimin-Lai-Sutherland ground state and SO$(n)$ matrix-product states using Bethe ansatz and nonlinear integral equation techniques.

Weaknesses

It is somewhat unclear which results are new and which are already known in the existing literature. (See also the report below.)

Report

In this paper, the authors introduce a class of non-Cardy conformal boundary states in the SU$(n)_1$ Wess-Zumino-Witten (WZW) theory by exploiting the conformal embedding Spin$(n)_2$ $\subset$ SU$(n)_1$. These states can be constructed by applying topological defect lines (TDLs) associated with the Spin$(n)_2$ theory. The authors further demonstrate that the ground states of certain Affleck-Kennedy-Lieb-Tasaki spin chains provide lattice regularizations of these conformal boundary states. In addition, they extract the Affleck-Ludwig boundary entropy by computing the overlap between the ground state of the SU$(n)$ Uimin-Lai-Sutherland model and the SO$(n)$ matrix-product state using Bethe ansatz techniques.

I find the results very interesting, and the manuscript appears to be free of noticeable errors. I am particularly impressed by the way in which the authors extract the Affleck-Ludwig boundary entropy through the analysis of the nonlinear integral equations. While similar methods have been employed in exact computations of the g-function in integrable field theories, their application to lattice models in this context is both original and insightful. I therefore believe that the manuscript merits publication in SciPost. Nevertheless, I would like to ask the authors to address the following issues before I can make a final recommendation.

  1. It is somewhat unclear which results are new and which are already known in the existing literature. I recommend that the authors clearly distinguish their original contributions from previously established results.

  2. Related to the above point: in Sec. 2, many formulas are presented without citation. For instance, the conformal dimensions of the primary fields in the Spin$(n)_2$ WZW model are listed without references. It would be helpful if the authors could provide appropriate references to the relevant literature (papers and/or textbooks).

  3. In Sec. 3, the authors take the critical Ising chain as an example and illustrate how the ground states of the gapped phases correspond to the three Cardy boundary states in the Ising CFT. I suggest that the authors provide appropriate references where this correspondence has been discussed.

  4. In Sec. 3, the phase diagram of the SO$(n)$ bilinear-biquadratic model is explained in words. It would, however, be helpful for the reader if the diagram were also presented schematically (for some fixed value of $n$).

  5. Eq. (25): The floor function is more commonly used to denote the integer part of a positive number. I also suggest that the authors provide a precise definition of the positive Bethe roots. (Although in the ground state the Bethe roots lie on the real axis, this is only mentioned later in Sec. 4.)

  6. In Appendix B, it is stated that the expansion of the transfer matrix $T(\lambda)$ in powers of $\lambda$ generates a family of mutually commuting operators, among which the ULS-chain Hamiltonian is included. Strictly speaking, however, the Hamiltonian is obtained from the logarithmic derivative of $T(\lambda)$.

Recommendation

Ask for minor revision

---

## Editorial Decision

awaiting_resubmission